# COPD Patients Exhibit Distinct Gene Expression, Accelerated Cellular Aging, and Bias to M2 Macrophages

**DOI:** 10.3390/ijms24129913

**Published:** 2023-06-08

**Authors:** Camila Oliveira da Silva, Jeane de Souza Nogueira, Adriana Paulino do Nascimento, Tatiana Victoni, Thiago Prudente Bártholo, Cláudia Henrique da Costa, Andrea Monte Alto Costa, Samuel dos Santos Valença, Martina Schmidt, Luís Cristóvão Porto

**Affiliations:** 1Laboratory of Histocompatibility and Cryopreservation, University of the State of Rio de Janeiro, Rio de Janeiro 20550-900, Brazil; camilavs162@gmail.com (C.O.d.S.); je.nogueira@gmail.com (J.d.S.N.); 2University Center of Valencia (UNIFAA), Rio de Janeiro 27600-000, Brazil; drikapn@yahoo.com.br; 3VetAgro Sup, University of Lyon, APCSe, 69280 Marcy l’Étoile, France; tatiana.victoni@gmail.com; 4Department of Thorax, University of the State of Rio de Janeiro, Rio de Janeiro 20550-900, Brazil; thiprubart@hotmail.com (T.P.B.); ccosta.uerj@gmail.com (C.H.d.C.); 5Tissue Repair Laboratory, University of the State of Rio de Janeiro, Rio de Janeiro 20550-900, Brazil; amacosta@uerj.br; 6Laboratory of Redox Biology, ICB, Federal University of Rio de Janeiro, Rio de Janeiro 21941-853, Brazil; 7Department of Molecular Pharmacology, University of Groningen, 9713 AV Groningen, The Netherlands; m.schmidt@rug.nl; 8Groningen Research Institute for Asthma and COPD, GRIAC, University Medical Center Groningen, University of Groningen, 9713 AV Groningen, The Netherlands

**Keywords:** COPD, HDAC, telomere, macrophage

## Abstract

COPD, one of world’s leading contributors to morbidity and mortality, is characterized by airflow limitation and heterogeneous clinical features. Three main phenotypes are proposed: overlapping asthma/COPD (ACO), exacerbator, and emphysema. Disease severity can be classified as mild, moderate, severe, and very severe. The molecular basis of inflammatory amplification, cellular aging, and immune response are critical to COPD pathogenesis. Our aim was to investigate EP300 (histone acetylase, HAT), HDAC 2 (histone deacetylase), HDAC3, and HDAC4 gene expression, telomere length, and differentiation ability to M1/M2 macrophages. For this investigation, 105 COPD patients, 42 smokers, and 73 non-smoker controls were evaluated. We identified a reduced HDAC2 expression in patients with mild, moderate, and severe severity; a reduced HDAC3 expression in patients with moderate and severe severity; an increased HDAC4 expression in patients with mild severity; and a reduced EP300 expression in patients with severe severity. Additionally, HDAC2 expression was reduced in patients with emphysema and exacerbator, along with a reduced HDAC3 expression in patients with emphysema. Surprisingly, smokers and all COPD patients showed telomere shortening. COPD patients showed a higher tendency toward M2 markers. Our data implicate genetic changes in COPD phenotypes and severity, in addition to M2 prevalence, that might influence future treatments and personalized therapies.

## 1. Introduction

Chronic obstructive pulmonary disease (COPD) is referred to by the Global Initiative for Chronic Obstructive Lung Disease (GOLD) as preventable and treatable with pulmonary and extrapulmonary symptoms that contribute to the severity of affected patients. COPD is characterized by airflow limitation that is not fully reversible. The prevalence of COPD in adults over 40 years old is around 9–10% in the world, being an important world cause of morbidity/mortality [1,2]. Although COPD might be caused by exposure to different toxic pollutants, the main risk factor in COPD development is smoking in more than 90% of cases [3]. However, only a minority, around 15% of smokers, develop COPD [4]. This factor, together with family susceptibility, indicates that COPD has an important genetic component involved in its genesis. Furthermore, COPD patients are highly heterogeneous in their clinical features, presentation, progression and are therefore classified into three specific and distinct phenotypes: overlapping asthma/COPD (ACO), exacerbator and emphysema with hyperinflation [5].

Many cell types like macrophages, epithelial cells, neutrophils, CD8 T lymphocytes, and, particularly in periods of exacerbation, eosinophils contribute to COPD inflammation [6]. Macrophages play a prominent role in COPD development and are increased five to ten times in bronchoalveolar lavage (BAL), lung parenchyma, and patient airways [7,8,9]. These cells seem to originate from circulating monocytes that can be distinguished into three subtypes: classic (CD14^++^CD16^−^), intermediaries (CD14^++^CD16^+^), and non-classical (CD14^+^CD16^++^) [10,11,12,13]. Classic subtypes may favor the differentiation of tissue macrophages into classically activated (M1) macrophages—highly competent in pro-inflammatory cytokines production, with microbicidal and cytotoxic activities—and nonclassical CD14^+^/CD16^++^ cells, which are more like resident tissue macrophages (M2), associated with Th2 response, resolution of inflammation, tissue repair, and reduction of pro-inflammatory cytokines [14]. Macrophages may also vary between M1 and M2 phenotypes to respond to the immunological needs arising from the pulmonary microenvironment [15]. The differentiation and activation of macrophages require the timely regulation of gene expression, which depends on the interaction of a variety of factors, including transcription factors and epigenetic modifications. Thus, dynamic changes of macrophages/monocytes in COPD may reflect and influence the stages of inflammation and remodeling of the pulmonary environment. In addition to cellular components, the inflammatory environment in COPD involves the presence of several pro-inflammatory mediators such as local and circulating lipids, cytokines, and chemokines, whose release is regulated by genetic transcription factors.

Exposure to noxious particles such as cigarette smoke alters airway epithelial barrier function and increases oxidative stress by promoting senescence and activation of pro-inflammatory pathways in airway epithelium. Increased oxidative stress also causes epigenetic changes such as DNA methylation and histone modification [16,17,18,19,20]. The underlying molecular mechanisms involved in the development and persistence of inflammation have not yet been fully understood. The balance of histone acetyltransferases (HATs) and histone deacetylases (HDACs) that promote histone acetylation and deacetylation, respectively, have been reported as influential elements in the transcription of pro-inflammatory genes [21,22].

Epigenetic factors play a critical role in chronic inflammation. Hypersecretion of airway mucus, decline in lung function, and exacerbation are serious pathophysiologic symptoms in COPD patients. EP300 can acetylate several proteins that regulate lung function and glandular secretion. EP300 was observed to inhibit MUC5AC gene expression in A549 cells. MUC5AC is a gel-forming mucin and is secreted at high levels in COPD [23]. EP300 has been the most studied protein associated with transcriptional activation of several genes in response to cell signaling. Increased activity and expression of EP300 was observed to be associated with different diseases, including pulmonary fibrosis and acute respiratory distress syndrome [24,25]. It was also observed that genetic deficiency and pharmacological inhibition of EP300 could abrogate pulmonary fibrosis both in vitro and in vivo [25,26].

Dysregulation of HDACs may act as a central point in the altered inflammatory state in the lungs of patients with COPD. Reduced HDAC activity in COPD is highlighted as an important mechanism for amplifying inflammatory gene expression in the disease. Marked reduction in HDAC activity in the lung parenchyma is correlated with disease severity and resistance to the anti-inflammatory effects of corticosteroids [27,28]. Tianwen Lai et al., observed that HDAC2 deficiency not only amplified airway inflammation, but also exacerbated the remodeling process and airway obstruction in patients with severe COPD [29]. HDAC2 expression and its enzymatic activity were significantly reduced in patients with COPD, a process correlated with excessive inflammation [30]. Decreased HDAC2 activity is associated with glucocorticoid resistance, increased oxidative stress, and increased production of pro-inflammatory mediators in alveolar macrophages of patients with COPD [31]. HDAC2 can induce IL-17 producing T cells. IL-17A and HDAC2 expression in the lung tissue of COPD patients was correlated with collagen deposition and bronchial wall thickening. The activation of HDAC and/or inhibition of IL-17A could prevent the development of airway remodeling by suppressing airway inflammation and modulating fibroblast activation in COPD [28,29,32].

HDAC3 inhibitors have been proposed as therapeutic targets to combat inflammation in COPD and asthma, since HDAC3 is a positive regulator of NF-κB mediated inflammation [33]. In macrophages stimulated by LPS/IFN-γ, selective inhibition of HDAC3 attenuated the NF-κB transcriptional activity, resulting in anti-inflammatory effects. Furthermore, HDAC3 siRNA-mediated down-regulation reduced the expression of pro-inflammatory genes IL-1β, IL-6, and IL-12b, corroborating the relevance of HDAC3 in the inflammation settings [34].

HDAC4 has been suggested as an important factor in pulmonary fibrosis being able to modulate extracellular matrix (ECM) production in pulmonary myofibroblasts [35,36,37]. An increased HDAC4 expression was observed in the diaphragm of COPD patients compared to controls [38].

From epigenome–wide studies in COPD, many differentially methylated genes have also been identified as being associated with COPD phenotypes [39].

Understanding the expression and activity profile of these enzymes (HAT and HDAC) in different COPD phenotypes may further enlarge our knowledge about COPD pathophysiology and may accelerate the development of specific targeted inhibitors, consequently leading to an improved direction and perspective of personalized therapy according to the patient COPD phenotype classification.

COPD is related to aging and has been linked to accelerated telomere wear [40]. It is already known that telomere shortening is associated with the risk of COPD and decreased lung function [41]. Leukocyte telomere length has been used as a biomarker of aging [42], which improves the treatment method and the effect of targeted therapies in patients with bone marrow failure and telomere shortening [43]. Several studies have evaluated telomere length in COPD context [18,44,45], but the relation between telomere length in different COPD profiles and severity has not been previously studied.

Our work demonstrates that a reduction of HDAC2 in mild, moderate, and severe COPD patients. In addition, a significant HDAC3 reduction in moderate and severe and HDAC4 increases in mild COPD was observed. A reduction in EP300 was specially observed in severe COPD. Emphysema and exacerbator present significant reduction in HDAC2 expression, and only emphysema showed HDAC3 reduction. Surprisingly, telomere shortening was detected in the smokers and COPD patients even in different age groups, as well as in all disease stages and ACO, exacerbator, and emphysema. Last, COPD patients demonstrated high tendency to express M2 markers (CD163 and CD206).

## 2. Results

### 2.1. COPD Patients Show Decreased Gene Expression of the EP300, HDAC2, and HDAC3 but an Increase in Gene Expression of HDAC4

To assess the role of histone acetylation and deacetylation enzymes in COPD, we evaluated by RT-PCR, EP300, HDAC2, HDAC3, and HDAC4 gene expression in circulating blood cells of the COPD patients, smokers, and healthy controls. We found a significant reduction in the expression of the EP300, HDAC2, and HDAC3 and an increase of the HDAC4 in COPD patients compared to smokers and healthy non-smoker controls (Figure 1a–d).

When we analyzed the COPD patients according to disease severity (mild, moderate, severe, and very severe), based on lung function by the spirometry test, a significant reduction of the EP300 gene expression was observed in severe COPD patients (Figure 1e), and a notable HDAC2 reduction in mild, moderate, and severe COPD patients (Figure 1f). Moreover, the HDAC3 gene expression was reduced only in moderate and severe COPD patients (Figure 1g), and an increase of the HDAC4 expression was observed only in mild COPD patients (Figure 1h).

Based on their clinical characteristics and symptoms, three main phenotypes were ascribed to COPD patients: overlapping asthma/COPD (ACO), exacerbator, and emphysema. Some individuals showing overlapping symptoms are difficult to classify in a specific phenotype and are therefore listed as unclassified. In the phenotypes analysis, we found that, except for the ACO, the EP300 gene expression is similar to the healthy controls and smokers (Figure 1i). A significant reduction in HDAC2 expression was evident in emphysema, exacerbator, and COPD unclassified (Figure 1j). Furthermore, only emphysema patients showed a significant reduction in HDAC3 expression (Figure 1k), and only COPD unclassified have a significantly increased expression in the HDAC4 gene (Figure 1l).

### 2.2. Cellular Aging, Revealed by Telomere Shortening, Is a Hallmark of COPD Patients

Relative telomere length was measured by RT PCR to evaluate the influence of the clinical disease condition on the total circulating blood cell aging. We observed that smokers and COPD patients showed a significant telomere shortening in relation to the control group (Figure 2a). Even separating in two distinct age groups (42–60 years and 61–84 years), smokers and COPD patients still showed more telomere shortening than the control group (Figure 2b,c).

When we evaluated between disease severity, smokers and mild, moderate, and severe COPD patients had significantly shorter telomeres than the non-smoking control group (Figure 2d). Among the different phenotypes, significantly shorter telomeres were observed among smokers and COPD patients of all phenotypes ACO, emphysema, and exacerbator (Figure 2e).

### 2.3. COPD Patients Present a Differentiation Trend to M2 Macrophage Profile

Macrophages are traditionally classified into M1 macrophages and M2 macrophages, where M1 macrophages are more pro-inflammatory and have cytotoxic properties, while M2 macrophages are anti-inflammatory and are involved in resolving inflammation and tissue repair processes [46,47]. We investigated, by flow cytometry, the expression of the CD38 and CD86 M1 markers, and CD163 and 206 M2 markers, in cultures of macrophages derived from human peripheral blood monocytes. Compared to controls, COPD patients express less CD38, regardless of whether or not they were stimulated with M1 growth factor GM-CSF and LPS (0.1 µg/mL), inflammatory stimulus (Figure 3a). We did not observe any difference between controls and COPD patients in the cellular expression of CD86 (Figure 3b).

Subsequently, when we evaluated CD163 and CD206 expression, markers of the M2 profile, there was a predominance of CD163 (Figure 3c) and CD206 (Figure 3d) in both culture conditions with MC-SF, M2 growth factor, stimulated or not with IL-4 (10 ng/mL), M2 stimulus. Representative histograms showing CD38, CD86, CD163, and CD206 expression are demonstrated in Appendix A.

## 3. Discussion

COPD is a heterogeneous disease in its clinical manifestation, progression, and lung lesions. It can be classified into four severities considering FEV1 (%predicted): mild (≥80%), moderate (50–79%), severe (30–49%), and very-severe (<30%) [2]. In addition, three main phenotypes are established, based on a single finding or through a combination of characteristics present in patients: COPD and asthma (ACO), exacerbator, and emphysema with hyperinflation.

The study aim was to investigate whether COPD patients, especially those with different phenotypes and severity, and smokers without the disease have specific genetic alterations in HAT-HDAC balance and telomeres, as well as alterations in the cellular profile of differentiated macrophages.

Inflammation in COPD is mediated by several regulatory mechanisms. At the molecular level, cellular signaling in inflammation is regulated by post-translational modifications (PTMs). The most studied PTMs are acetylations, in which an acetyl group is added to a lysine residue by a HAT enzyme or in which an acetyl group is removed by an HDAC enzyme.

EP300 is one HAT widely studied and important for the coactivation of several transcription factors, including NF-κB and activator protein-1 (AP-1). Increased acetylation of histones (H3/H4) and NF-κB by CBP/EP300 are associated with cigarette smoke-mediated release of pro-inflammatory cytokines [48,49,50]. Contradicting the expected, our results showed a reduction in EP300 gene expression, especially in patients with severe COPD, despite the heterogeneous distribution of our cohort participants. However, when we separated by phenotypes, there was no difference between the groups compared to smokers and non-smoker controls. We believe that this reduction, in severe cases, may be associated with exaggerated response control mechanisms that are occurring specifically in these patients. 

The reduced expression and activity of HDAC2 in peripheral blood mononuclear cells (PBMCs) from COPD patients compared to smokers and non-smokers is well described [27,51]. We observed that COPD patients have a decreased HDAC2 gene expression in whole peripheral blood, corroborating previous studies [52]. In addition, we also observed that in all disease severities and in the emphysema, exacerbator, and unclassified phenotypes, HDAC2 was decreased in both smokers and non-smoker controls, which may contribute to chronic inflammation with subsequent remodeling and airway obstruction. HDAC3 is identified as a positive regulator of NF-κB-mediated inflammation. HDAC3 inhibitors have been proposed as therapies to combat inflammation in COPD [33]. Our results were different from those studies, as there was a decrease in HDAC3 expression only in patients with the emphysema phenotype and with moderate and severe severity. Increased levels of HDAC4 are described in the diaphragm of COPD patients compared with controls [38]. According to our results, an increase in HDAC4 is also present in peripheral blood cells of COPD patients with mild severity and among the unclassified. An in vivo and in vitro study using HDAC4 inhibitors suggested an anti-inflammatory role for this enzyme in patients with the emphysema phenotype. Furthermore, HDAC4 are important in pulmonary fibrosis, being able to modulate extracellular matrix (ECM) production in pulmonary myofibroblasts [36,37,53].

Decreased expression and enzymatic activity of HDAC2 in COPD patients may contribute to premature aging [31]. It is already known that cigarette smoke is a major factor in telomere shortening [54,55]. Studies that evaluated the relationship between telomere length and COPD showed shorter telomeres in subjects with COPD than in smokers and healthy subjects [56,57]. Additionally, telomere shortening in leukocytes in a large population-based cohort of smokers was already reported [54], in line with our results in which shorter telomeres were detected in smokers in the same proportion of COPD patients compared to non-smoking controls. Furthermore, our results suggest that smoking might be a stronger factor in telomere shortening in COPD disease. When we analyze the disease severity, we see that this shortening is maintained for all severities except for the very severe case; due to the sample size, however, it was not possible to fully confirm these data. All the phenotypes showed reduced telomeres when compared to the non-smoking control group.

As already mentioned, the macrophage is one of the main cells that acts in COPD, and its origin comes from circulating monocytes. It has been described that patients with severe COPD have an increase in non-classical monocytes, suggesting the development of a population of monocytes with a pre-M2 phenotype, more likely to differentiate into M2 macrophages in lung tissue after analysis of type markers M2, such as CD163 and CD206 [58,59,60]. Higher levels of M2 cytokines (TARC/CCL17 and MDC/CCL2) were observed in IL-4-stimulated monocytes from COPD patients than in cells from healthy donor. In addition, a significant increase in the frequency of intermediate monocytes was observed in COPD patients compared to healthy controls in a previous study of our group [61]. Regarding disease severity, we found an intermediate increase in monocytes in mild, moderate, severe, and very severe COPD patients. Exacerbating COPD patients were the only ones who showed an increase in intermediate and non-classical monocytes (Appendix A).

It is known that CD38 and CD86 proteins are upregulated in M1-type macrophages and are considered M1 markers. The CD38 marker is a cell surface glycoprotein and extracellular enzyme required for activation and proliferation of immune cells [62]. Marker CD86 is an integral membrane glycoprotein that is expressed on the surface of activated T lymphocytes, B lymphocytes, and antigen presenting cells (APC) [63]. The CD163 and CD206 proteins are two markers used to evaluate the M2 profile [59,64]. CD163 is a receptor of the hemoglobin/haptoglobin complex [65,66], and CD206 (mannose receptor) is a receptor involved in the phagocytosis of bacteria and fungi [67]. Our results demonstrate an increase in the expression of M2 markers CD163 and CD206 and a decrease of CD38 (M1 marker) in monocyte-derived macrophages from PBMC of the COPD patients. Furthermore, previous results from our group showed a decrease in CD86^+^ macrophages and an increased number of CD206^+^ cells in the BAL of mice exposed to cigarette smoke for 14 days [68].

The epigenetic modifications of macrophage polarization are concentrated in the post-translational modifications of histones. Both histone methylation and acetylation are important for the polarization of M1 and M2. These epigenetic macrophage modifications were also demonstrated in AMs from COPD patients, with increased basal expression of inflammatory markers that persist even after prolonged ex vivo culture [69]. It was also observed that HDAC3 acts as a brake on IL-4-induced M2 polarization [70].

Another study observed the phenotypic differentiation of macrophage subpopulations in the small airway wall and in the airway lumen of patients with COPD. In the small airways there was a predominance of M1 macrophages, while in BAL there was a change to the M2 profile. In addition, an increase in M2 cytokines in the BAL of patients and an increase in arginase enzyme in the airway wall mucosa, including the epithelium, sub epithelium and alveolar septa, was observed, which could suggest the presence of the M2 macrophage in these regions [71].

Malyshev et al. raised the hypothesis of an M1 and M2 interconversion forming the M3 phenotype [72]. Given the complexity of the microenvironment in COPD, a bare classification of the macrophage into M1 or M2 would be arbitrary. Macrophages are no longer restricted to two extreme phenotypes but can have different phenotypes, which include pro-inflammatory, pro-fibrotic, pro-tumor, and anti-inflammatory profile.

In fact, studies on the severity and different phenotypes of the disease are extremely relevant for the development of treatments that reflect the heterogeneity of the disease, reinforcing the importance of this study. In conclusion, our data collectively indicate that COPD patients showed a reduction in EP300, HDAC2, and HDAC3 gene expression and an increase in HDAC4. The HDAC2 reduction can lead to DNA damage and promote premature aging. We found that COPD patients have more telomere length shortening than healthy individuals of the same age group. Finally, we observed that COPD patients present a trend differentiation to the M2 profile (CD163 and CD206) when induced in culture (Figure 4).

## 4. Materials and Methods

### 4.1. Study Population

This study was approved by the Ethics Committee of the Pedro Ernesto University Hospital/UERJ, (CAAE): 18267619.7.0000.5259 and opinion number: 3.655.838. All recruited subjects signed the Informed Consent Form. The study population is composed of patients coming from the COPD and Smoking Outpatient Clinics of the Pneumology Service—UERJ. Healthy individuals were selected among employees of the institution.

Volunteers eligible for the study were divided into three groups: the first corresponds to the healthy control group, including non-smokers with no evidence of lung disease on spirometry, aged 40 years; the second group corresponds to smokers with no evidence of lung disease on spirometry, aged 40 years; the third group is composed by patients with COPD, aged 40 years, diagnosis of the disease confirmed by spirometry, in addition to a smoking history of more than 20 packs per year. Patients with COPD presented FEV1 (forced expiratory volume in one second)/forced vital capacity (FVC) ratio below 0.7 after the use of salbutamol 400 µg.

COPD patients with different phenotypes and severities were divided according to the criteria mentioned below.

For the different disease stages, patients were classified as mild, FEV1 ≥ 80% predicted; moderate, 50% ≤ FEV1 < 80% predicted; severe, 30% ≤ FEV1 < 50% predicted; and very severe, FEV1 < 30% of predicted. Spirometric classifications of COPD severity were described by GOLD 2023 [2].

For the different COPD phenotypes, patients with COPD were subdivided according to six clinical criteria that define the three different phenotypes for this study: history of atopy, number of exacerbations in the last 12 months, response to the bronchodilator test, diffusion values for percent CO, percent of total lung volume, and percentage of residual volume. Individuals who did not have the characteristics of a phenotype or who had characteristics related to more than one phenotype were included as unclassified. The characteristics of COPD patients, smokers, and recruited controls are described by Bartholo et al. [73]. A total of 105 patients with COPD, 42 smokers without COPD, and 73 healthy controls were recruited. The demographic characteristics of the study population are summarized in Table 1.

### 4.2. Obtaining Monocytes from Human Peripheral Blood

For this step, whole blood (two EDTA tubes) was collected from the volunteer donors. Subsequently, this blood was diluted 1:1 in PBS1× containing 2 mM EDTA (Invitrogen Eugene, OR, USA) and added slowly into a tube containing 10 mL of Ficoll-Paque Plus (Ge Healthcare Life Sciences, Little Chalfont, UK). The tube was centrifuged at 500× *g* for 30 min at 20 °C. The leucocytes layer was collected with a pipette, transferred to a new tube, and diluted with PBS1× (+EDTA) until 12 mL. A centrifugation was performed 500× *g* for 10 min at 20 °C to wash cells. After discarding the supernatant, the cell pellet was resuspended in 1 mL of RPMI 1640-Glutamax complete culture medium (Sigma-Aldrich, St. Louis, MO, USA) with 1% penicillin and 0.5% streptomycin (Invitrogen Eugene, OR, USA). Cells were counted in the Neubauer chamber and seeded in 24-well plates at a concentration of 2 × 10^6^ cells/mL/well.

### 4.3. Cell Culture and Stimulus

Cells were seeded in 24-well plates and incubated at 37 °C with 5% CO_2._ After 1 h of incubation, the RPMI 1640-GlutaMAX medium was removed to eliminate non-adherent cells. In this protocol, mostly adherent cells are predominantly monocytes. These cells were cultivated for five days in complete culture medium RPMI 1640-Glutamax containing 1% penicillin, 0.5% streptomycin, and 10% fetal bovine serum and supplemented with 50 ng/mL of GM-CSF (R&D System, Minneapolis, MN, USA) in the cultures for the M1 profile, or 10 ng/mL M-CSF (R&D System, Minneapolis, MN, USA) in the cultures for the M2 profile.

At day four, the culture medium was removed, then the differentiated cells were deprived of GM-CSF and M-CSF for 24 h and stimulated with LPS (0.1 µg/mL) (*E. coli* 055: B5, Sigma-Aldrich, St. Louis, MO, USA) or IL-4 (10 ng/mL) (R&D System, Minneapolis, MN, USA) for 24 h before being retrieved for analysis. LPS and IL-4 stimuli were used as inducers (positive controls) of each profile in the culture. In conditions with LPS (inducer of the M1 profile), markers CD38 and CD86 should be expressed, whereas with IL4 (inducer of the M2 profile), markers CD163 and CD206 should be expressed.

### 4.4. Characterization of the Macrophage Profile

After five days, with the aid of a 200 µL tip, the differentiated cells were gently detached from the plate and collected. After washing with 4 mL of PBS1×, the cells were resuspended in 1 mL of PBS1× and counted in a Neubauer chamber.

The cell suspensions were distributed in a U bottom plate, centrifuged, and, subsequently, the cell pellet incubated in the dark with the surface antibodies: anti-CD38-FITC, anti-CD86-PE Cyanine5, anti-CD163-PE Cyanine7, and anti-CD206-PE (Invitrogen) in 100 µL PBS1× (+10% FCS) for 20 min. Next, 150 µL of PBS1× (+10% FCS) was added, and the cells were centrifuged in a 1700 rpm for 6 min and resuspended in 300 µL of PBS1× in cytometry tubes. The samples were acquired in a Gallios cytometer (Beckman Coulter) and analyzed using the Flowjo (BD, Franklin Lakes, NJ, USA) program.

### 4.5. RNA Extraction

Approximately 3 mL of whole blood was collected into a Tempus tube (Applied Biosystems, Woburn, MA, USA) containing a stabilizing reagent that lyses blood cells. This reagent inactivates cellular RNases and selectively precipitates RNA; genomic DNA and proteins remain in solution. For RNA extraction, a kit was used: Tempus Spin RNA Isolation Kit (Applied Biosystems, Woburn, MA, USA). In summary, approximately 400 mL of the resuspended RNA in the purification filter was added and centrifuged for 30 s at 16,000× *g*. After centrifugation, the purification filter was washed three times with wash buffers from the kit. Then, 100 μL of ribonucleic acid was added to the purification filter, incubated for 2 min at 70 °C, and centrifuged for 30 s at 16,000× *g*. The extracted RNA samples were quantified, the purity evaluated through the absorbance spectrum in the NanoDrop device (Thermo Scientific, Wilmington, DE, USA), and the samples stored at −80 °C until use.

For the synthesis of single-stranded cDNA from total RNA, the High-Capacity cDNA Reverse Transcription kit (Applied Biosystems, Woburn, MA, USA) was used for reverse transcription in a thermocycler. We used the following parameters in the thermal cycler: step one at 25 °C for 10 min; step two at 37 °C for 120 min; step three at 85 °C 5 min; and step four at 4 °C. The cDNA obtained was stored at –80 °C until gene expression assay performs.

### 4.6. Analysis of HAT and HDAC Expression

For the evaluation of gene expression, TaqMan Gene Expression Assay probes (Applied Biosystems) were used for the genes: HDAC2 (Hs00231032_m1), HDAC3 (Hs00187320_m1), HDAC4 (Hs01041638_m1), EP300 (Hs00914223_m1), GAPDH (Hs02758991_g1) (reference gene), and TaqMan Master Mix II with UNG (Applied Biosystems) for the real-time PCR reaction in the StepOne Plus equipment (Applied Biosystems, Woburn, MA, USA), respectively.

PCR reactions were set up in a final volume of 20 µL/sample. For the reaction, we add 10 μL of the TaqMan^®^ Universal Master Mix II (2×) (Applied Biosystems, Woburn, MA, USA), 1 μL TaqMan^®^ Gene Expression Assay (20×) (Applied Biosystems, Woburn, MA, USA), and 9 μL of the cDNA template diluted in DNase and RNase free ultrapure water. The reading was performed in StepOne Plus (Applied Biosystems, Woburn, MA, USA) using the following settings for target and reference genes: UNG incubation 50 °C for 2 min; polymerase activation 95 °C 10 min, denaturation 95 °C for 15 s; and annealing and extension (40 cycles) at 60 °C for 1 min. The analysis was performed in duplicates, and the Ct (cycle threshold) data were plotted and analyzed by the 2^−ΔΔCt^ (Livak’s method).

### 4.7. DNA Extraction

Whole peripheral blood (200 µL) was used for genomic DNA extraction. A commercial Biopur mini spin plus extraction kit was used, following the manufacturer’s instructions. Briefly, 25 µL of proteinase K and 200 µL of peripheral blood were added to a microtube. Then 200 µL of Lysis S buffer was added and vigorously homogenized in vortex for 10 to 20 s. The microtubes were incubated at 56 °C for 15 min. An amount of 210 µL of ethanol (96–100%) was added, and the samples were transferred to the spin S microtube and centrifuged for 1 min at 11,000× *g*. After centrifugation, the microfilter tube was washed with the wash buffers from the SI and SII kit. Next, the microfilter tube was placed in an S elution microtube, and 200 µL of S elution buffer previously heated to 56 °C was added. The buffer was dispensed directly onto the silica membrane, incubated for 1 min at room temperature, and centrifuged for 1 min at 11,000× *g*. The DNA were quantified, and the sample purity evaluated through the absorbance spectrum in the NanoDrop (Thermo Scientific, Wilmington, DE, USA) and stored at −80 °C.

### 4.8. Relative Quantification of Telomere Length by Real-Time PCR

Telomere length was evaluated using the real-time PCR technique with the comparative CT method (ΔΔCT). Quantification of a region of telomeric DNA was performed, compared to a region of the reference gene, β-globin.

The following primers [74,75,76] were used for the reaction:

Tel 1b: 5′ CGG TTT GTT TGG GTT TGG GTT TGG GTT TGG GTT TGG GTT 3′

Tel 2b: 5′ GGC TTG CCT TAC CCT TAC CCT TAC CCT TAC CCT TAC CCT 3′

HBB1: 5′ GCT TCT GAC ACA ACT GTG TTC ACT AGC 3′

HBB2: 5′ CAC CAA CTT CAT CCA CGT TCA CC 3′

PCR reactions were set up in a final volume of 20 µL/sample. For the telomere reaction, the following were used: 10 μL SYBR^®^ Green PCR Master Mix (Applied Biosystems, Woburn, MA, USA); 1.2 μL of tel 1b primer (10 μM); 1.8 μL of tel 2b primer (10 μM); 2 μL of DNA; and 5.0 µL of DNase and RNase free ultrapure water. For the β-globin reaction, we used 10 μL SYBR^®^ Green PCR Master Mix (Applied Biosystems, Woburn, MA, USA), 0.8 μL of HBB1 primer (10 μM), 0.8 μL of HBB2 primer (10 μM), 2 μL of DNA, and 6.4 μL of DNase and RNase-free ultrapure water. The reading was performed in StepOne Plus (Applied Biosystems, Woburn, MA, USA).

The target and reference genes were run on separate plates with different settings. For telomeres, we used the following parameters: denaturation at 95 °C for 10 min, annealing and extension (40 cycles) at 95 °C for 15 s and 54 °C for 2 min; for β-globin, denaturation at 95 °C for 10 min, annealing and extension (40 cycles) at 95 °C for 15 s and 58 °C for 1 min. In both cases, melting curves were performed at a temperature of 60 °C. The analysis was performed in duplicates, and the Ct (cycle threshold) data were plotted and analyzed by the 2^−ΔΔCt^ (Livak’s method). 

### 4.9. Statistical Analyzes

Results are expressed as mean ± standard error of mean. The normality of all data was tested using the Shapiro-Wilk test. Data were analyzed using one-way analysis of variance (ANOVA), followed by the Tukey and Bonferroni posttests. Student’s *t*-test and ANOVA test were used to assess differences in telomere length and the different variables analyzed. Last, *p* values < 0.05 were considered statistically significant. All data were plotted in graphs using GraphPadPrism software version 6.0 (San Diego, CA, USA).

## Figures and Tables

**Figure 1 ijms-24-09913-f001:**
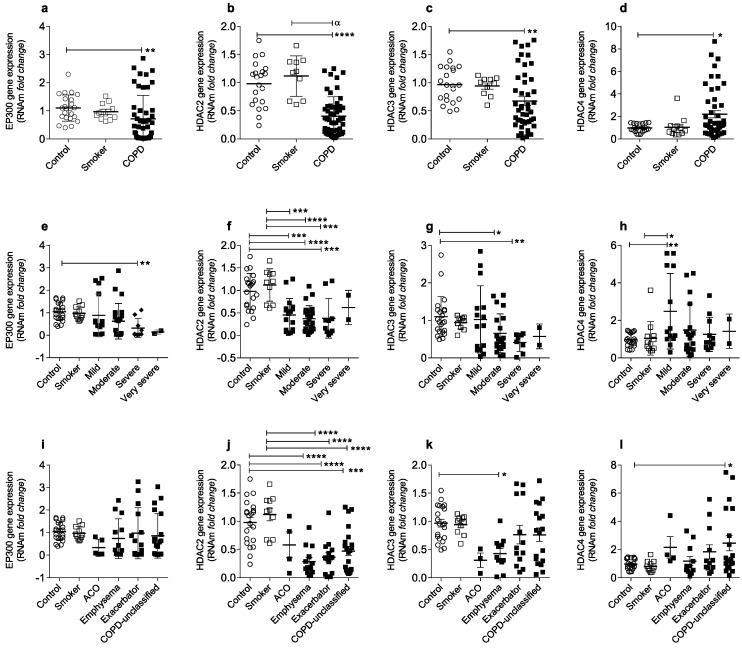
Decreased EP300, HDAC2, and HDAC3 gene expression and an increase HDAC4 gene expression in COPD patients. Expression of the EP300, HDAC2, HDAC3, and HDAC4 gene in whole peripheral blood cells of healthy non-smoking controls (*n* = 23), smokers (*n* = 13), and COPD patients (*n* = 56) (**a**–**d**); non-smoking controls (*n* = 23), smokers (*n* = 13), mild (*n* = 18), moderate (*n* = 24), severe COPD (*n* = 11), and very severe (*n* = 3) (**e**–**h**); non-smokers (*n* = 23), smokers (*n* = 13), ACO (*n* = 5), emphysema (*n* = 14), controls exacerbator (*n* = 14), and COPD-unclassified (*n* = 23) (**i**–**l**). Statistics were performed using a one-way analysis of variance (ANOVA), followed by a Bonferroni test. Data represent the mean ± standard error of the mean. White circles represent healthy non-smoking controls; white squares represent smokers; black squares represent COPD patients. Significance of *p*-values: * *p* < 0.05, ** *p* < 0.01, *** *p* < 0.001 and **** *p* < 0.0001 compared to control, α 0.05 compared to smoker.

**Figure 2 ijms-24-09913-f002:**
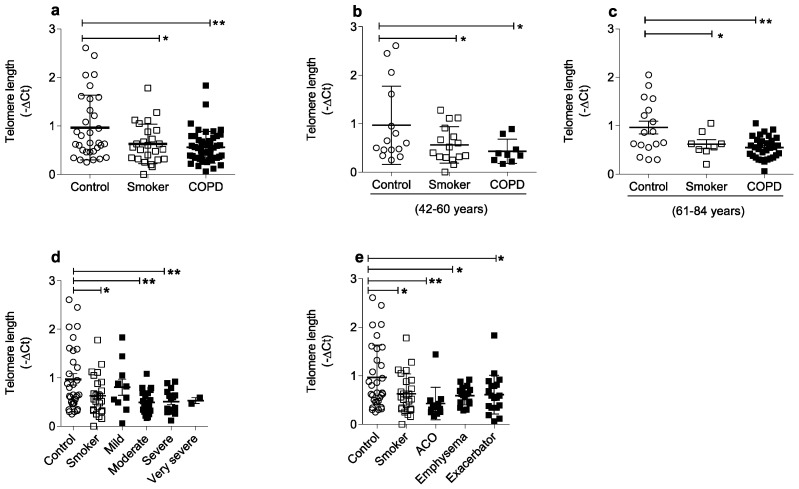
Smokers and COPD patients of all phenotypes showed a significant telomere shortening. Telomere length in non-smoking controls (*n* = 40), smokers (*n* = 29), and COPD patients (*n* = 66) (**a**); 42–60 years old non-smoking controls (*n* = 23), smokers (*n* = 20), COPD patients (*n* = 17) (**b**); 61–84 years old non-smoking controls (*n* = 17), smokers (*n* = 9), COPD patients (*n* = 49) (**c**); mild, *n* = 13 (61–84 years, *n* = 13); moderate, *n* = 30 (42–60 years old *n* = 13; 61–84 years old, *n* = 17); severe, *n* = 19 (42–60 years old, *n* = 7; 61–84 years old, *n* = 12); and very severe, *n* = 3 (42–60 years old, *n* = 1; 61–84 years old, *n* = 2) (**d**). ACO, *n* = 14 (42–60 years old, *n* = 7; 61–84 years old, *n* = 7); emphysema, *n* = 25 (42–60 years old *n* = 6; 61–84 years old *n* = 19); exacerbator, *n* = 27 (42–60 years, *n* = 4; 61–84 years, *n* = 23) (**e**). Statistics were performed using a one-way analysis of variance (ANOVA), followed by a Bonferroni test. White circles represent healthy non-smoking controls; white squares represent smokers; black squares represent COPD patients. Data represent the mean ± standard error of the mean. Significance of *p*-values: * *p* < 0.05, ** *p* < 0.01 compared to control.

**Figure 3 ijms-24-09913-f003:**
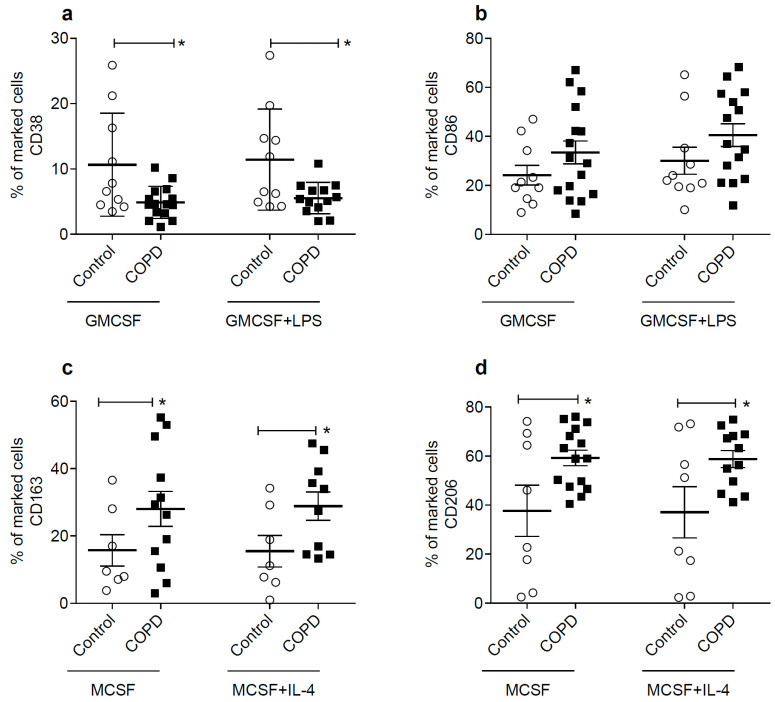
COPD patients present lower CD38 expression, as well as higher CD163 and CD206 expression. Mean percentage of cells positive for CD38 control (*n* = 10) and COPD patients (*n* = 15) (**a**); CD86 control (*n* = 10) and COPD patients (*n* = 15) (**b**); CD163 control (*n* = 7) and COPD patients (*n* = 10) (**c**); and CD206 control (*n* = 8) and COPD patients (*n* = 12) (**d**) in cultured macrophages derived from human peripheral blood monocytes were determined by flow cytometry. Statistics were performed using a one-way analysis of variance (ANOVA), followed by a Bonferroni test. Data represent the mean ± standard error of the mean. White circles represent healthy non-smoking controls; black squares represent COPD patients. Significance of *p*-values: * *p* < 0.05 compared to control.

**Figure 4 ijms-24-09913-f004:**
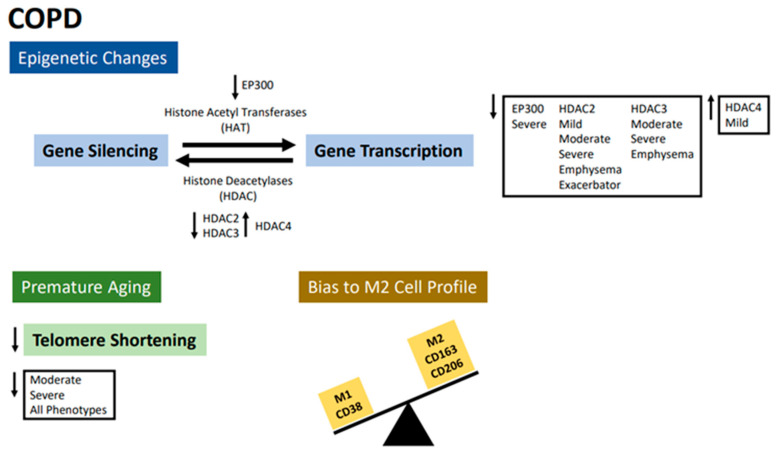
Our data suggest that COPD patients are characterized by genetic alterations in phenotypes and disease severity, as well as changes in macrophage profile.

**Table 1 ijms-24-09913-t001:** The demographic characteristics of the study population.

Variable	Control (*n* = 73)	Smoker (*n* = 42)	COPD (*n* = 105)	ACO (*n* = 14)	Emphysema (*n* = 25)	Exacerbator (*n* = 27)	Unclassified (*n* = 39)
Gender Male/Female	25/48	15/27	55/50	7/7	12/13	19/8	17/22
Age (Years)	56 ± 1.13 (42–84)	58 ± 1.10 (42–76)	67 ± 0.81 (42–87)	60 ± 2.10 (42–72)	66 ± 1.56 (54–79)	67 ± 1.55 (53–83)	71 ± 1.19 (55–87)
Smoking (pack/year)	0	39 ± 1.92	47 ± 2.48	60 ± 7.31	44 ± 2.8	48 ± 5.55	45 ± 5.94
FEV_1_, liter	2.58 ± 0.08	2.38 ± 0.081	2.61 ± 0.95	1.68 ± 0.14	4.6 ± 3.35	1.56 ± 0.13	1.58 ± 0.16
FEV, %	98.99 ± 2.12	92.42 ± 1.59	35.57 ± 2.15	68.28 ± 4.09	50 ± 4.58	58.42 ± 3.99	67.57 ± 5.12
FEV_1_, liter	3.19 ± 0.10	3.02 ± 0.12	2.86 ± 0.008	3 ± 0.26	2.61 ± 0.17	3.06 ± 0.18	2.83 ± 0.17
FVC, %	97.22 ± 2.03	93.14 ± 1.95	93.48 ± 2.2	100 ± 5.08	87.88 ± 3.81	91.95 ± 4.83	99.07 ± 5.66

Abbreviations: COPD, chronic obstructive pulmonary disease; ACO, overlapping asthma/COPD; FVC, forced vital capacity; FEV_1_, forced expiratory volume in 1 s. Values are expressed as mean ± standard error of the mean.

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
