# Peer review of "COPD Patients Exhibit Distinct Gene Expression, Accelerated Cellular Aging, and Bias to M2 Macrophages"

_ijms, 2023, doi:10.3390/ijms24129913_

Round 1
Reviewer 1 Report
The authors investigate gene expression in a cohort of controls, smokers and COPD. While the results are interesting they are presented simply with no detailed analysis or linkage between the various outcomes. I have several questions;
1. Were the COPD phenotypes associated with COPD disease severity?
2. Were any of the outcomes associated with smoking pack year history?
3. Why was telomere length assessed in 42-60 vs 61-84 year old groups? This seems rudimentary. Why was it not assessed in relation to age with status (eg control, smoker or COPD or pack year history) as an independent variable?
4. The results are confusing when assessed in terms of COPD severity and phenotype overlap. Similarly, the description of EP300, HSAC2/3/4 is hard to interpret. It would be more informative if a multivariable regression was used to show associations.
5. Should severity be used instead of graveness
Author Response
- Were the COPD phenotypes associated with COPD disease severity?
R: We analyze COPD phenotypes and severity as distinct evaluations. The different COPD phenotypes were determinate according to six clinical features: atopy history, number of exacerbations in the last 12 months, bronchodilator test response, diffusion values for percentage of CO, percentage of total lung volume, and percentage of residual volume. The disease severity, patients were classified as mild, FEV1≥80% predicted; moderate, 50%≤FEV1<80% predicted; severe, 30%≤ FEV1<50% predicted; and very severe, FEV1<30% of predicted. Individuals who did not present specific phenotype’s feature or who presented more than one specific phenotype’s feature were included in the unclassified group. These two classifications could be analyzed associated (Ex: studying the severities within each phenotype), but the number of patients would compromise the analysis, so we choose for individual approach where we also reached our goal of to study genetic and immunological aspects of COPD between clinically heterogeneous patients.
- Were any of the outcomes associated with smoking pack year history?
R: We do not have information of the clinical outcomes of COPD patients, therefore, associations with the annual number of smoking packs were not done.
- Why was telomere length assessed in 42-60 vs 61-84 year old groups? This seems rudimentary. Why was it not assessed in relation to age with status (eg control, smoker or COPD or pack year history) as an independent variable?
R: Telomere measurements are used as a helpful tool to predict biological age. It is well established in the literature that advanced age is a critical factor to telomere shortening. In this study, COPD patients, smokers, and controls have age group over 40 years old (spanning ages 42 to 84, which could also cause a variation in the loss of telomeres). We chose to analyze two main age groups (middle-aged individuals, from 45 to 59 years old and elderly, over 60 years old) to investigate whether the reduction observed when analyzing total N could also be associated with the disease, COPD, as a factor that helps to accelerate peripheral cell aging. Even after division in two distinct age rates, similar results were observed, corroborating the previously data of whole sample size (Figure 2a). Furthermore, our goal was to show the statistical differences between the groups highlighted the unexpected result of smokers.
- The results are confusing when assessed in terms of COPD severity and phenotype overlap. Similarly, the description of EP300, HSAC2/3/4 is hard to interpret. It would be more informative if a multivariable regression was used to show associations.
R: Our article is the first to demonstrate gene expression profile results in patients with distinct clinical features, that why we opted to demonstrate separately and to evaluate if there is a more specific impairment in any of the proposed phenotypes or severity.
- Should severity be used instead of graveness.
R: We replaced the word graveness by severity in the entire manuscript.
Reviewer 2 Report
The manuscript of da Silva et al. presents a set of analyses of PBMC from COPD patients indicating differential expression of several genes and gene groups in COPD patients vs. controls, which are not properly described (see below). This is a very modest and mostly descriptive set of data with little/no functional insight. It still deserves to be published to become available to the broader COPD community after the following serious deficiencies in presentation are corrected. 1. More information about the function(s) of EP300 and especially HDAC family and how it may be related to COPD pathogenesis should be given in Introduction. Otherwise, it's difficult to understand authors' rationale. 2. Since it is stated that all members of COPD panel have been smokers (lines 321-323: "the third group is composed by patients with COPD aged 40 years, diagnosis of the disease confirmed by spirometry, in addition to a smoking history of more than 20 packs per year"), all the key comparisons should be done for non-COPD smoker vs. COPD smoker cohorts and not vs. healthy volunteers. This has a major impact on paper conclusions 3. It is very clear from Figs. 1a-d that expression of all genes tested by authors in COPD patients is highly heterogeneous. This should be stated, and it also makes Figs. 1e-h actually more important. This, in turn, means that only findings for HDAC2 withstand a reasonable degree of scrutiny, while other differences look fairly random even if they score statistical significance here and there vs. healthy controls (notably, authors elect to show error bars as SEM and not by SD, which is more appropriate). Panels 1i-1m only support this conclusion since only HDAC2 decrease is uniformly observed across different COPD phenotypes and other differences look to be random (with the possible exception of HDAC3 decrease in emphysema patients but, again, only vs. healthy controls, which is a moot point). 4. Thus, telomere length as reported in Fig. 2 is clearly affected by smoking and not by COPD and should be reported as such. 5. Rationales of using strong macrophage stimulators such as LPS (and also IL-4) in Fig 3 is not described and, in fact, it makes no difference and can be safely dropped. And since there is no proper control in this Figure (non-smokers are compared to smoking COPD patients), it has a very limited value, not to mention that number of patients stated in the Figure legend (10 normal and 16 COPD ones) does not coincide with symbols seen in panels c and d. 6. Additional major limitations of this study looking only at PBMC should be noted in Discussion. 7. Accurate read-through and extensive set of language corrections is needed to avoid the use of incorrect terms or words (e.g., "in all disease degrees" on lines 119-120 instead of "disease stages" or "levels of disease progression" or, similarly, "disease graveness" on line 131 and in many other places instead of "disease severity" or "so called then as unclassified", line 141 instead of " and are therefore listed as unclassified" or " telomere length was performed", line 159, instead of "was measured"; there are many, many more examples of this).
Please see my general comments. The manuscript is readable but has many instances where a much more appropriate term/wording should be used. There are a few grammar mistakes as well + missing words (e.g., ", regardless they were stimulated" instead of "regardless whether they were stimulated," line 188).
Author Response
- More information about the function(s) of EP300 and especially HDAC family and how it may be related to COPD pathogenesis should be given in Introduction. Otherwise, it's difficult to understand authors' rationale.
R: We appreciate the suggestion and, as requested, we added more information about function(s) of the EP300 and HDAC2 and how this may be related to the pathogenesis of COPD, this information was included in the introduction of the manuscript. Basically, the great impact of these enzymes is on the induction and propagation of the inflammatory process, which consequently produced a series of other bystander effects in the lung tissue.
- Since it is stated that all members of COPD panel have been smokers (lines 321-323:"the third group is composed by patients with COPD aged 40 years, diagnosis of the disease confirmed by spirometry, in addition to a smoking history of more than 20 packs per year"), all the key comparisons should be done for non-COPD smoker vs. COPD smoker cohorts and not vs. healthy volunteers. This has a major impact on paper conclusions
R: Since the beginning of the study, smokers were included as an investigational group, as well as COPD patients due to smoking being a critical factor in COPD. Regarding gene expression, we found no differences between the control group of healthy individuals and smokers, suggesting that epigenetics does not seem to be affected by smoking. Surprisingly, the telomere shortening found in smokers, in addition to being associated with age, is a result that requires further studies to understand how the components of cigarette smoke act in cellular biology to the point of promoting this reduction. We performed expression analyzes of the M1 and M2 profile markers of differentiated macrophages in culture using the healthy control as a basic comparative group instead of the smokers who could induced a bias, similar to that of telomeres, probably due to the cumulative effects of cigarette smoke. Furthermore, the recruited COPD patients had different smoking periods, which could lead to considerable heterogeneity in the results. We didn't get to evaluate smokers in this profile studies but would be interesting to know the macrophage profile bias in this group.
- It is very clear from Figs. 1a-d that expression of all genes tested by authors in COPD patients is highly heterogeneous. This should be stated, and it also makes Figs. 1e-h actually more important. This, in turn, means that only findings for HDAC2 withstand a reasonable degree of scrutiny, while other differences look fairly random even if they score statistical significance here and there vs. healthy controls (notably, authors elect to show error bars as SEM and not by SD, which is more appropriate). Panels 1i-1m only support this conclusion since only HDAC2 decrease is uniformly observed across different COPD phenotypes and other differences look to be random (with the possible exception of HDAC3 decrease in emphysema patients but, again, only vs. healthy controls, which is a moot point).
R: We mentioned in the fourth paragraph of the discussion the heterogeneity of the group of COPD patients. We know that in our study there is a limitation of the sample number per group to generate large conclusions. Our isolated significant results, comparing some of the phenotypes or severities with healthy controls, are suggestive data that can contribute to the understanding of the clinical presentation of the disease and possible factors, such as genetic and inflammatory alterations, which may be implicated in their origins. Consequently, these reported data may help guide future individualized treatments for specific patients and understand the resistance to certain medicine.
- Thus, telomere length as reported in Fig. 2 is clearly affected by smoking and not by COPD and should be reported as such.
R: In the sixth paragraph of the discussion, we added a sentence highlighting that smoking should be a determining factor in telomere shortening in COPD.
- Rationales of using strong macrophage stimulators such as LPS (and also IL-4) in Fig 3 is not described and, in fact, it makes no difference and can be safely dropped. And since there is no proper control in this Figure (non-smokers are compared to smoking COPD patients), it has a very limited value, not to mention that number of patients stated in the Figure legend (10 normal and 16 COPD ones) does not coincide with symbols seen in panels c and d.
R: We add the rationale in the end of the item “4.3 Cell culture and stimulus” of the Materials and Methods:
LPS and IL-4 stimuli were used as inducers (positive controls) of each profile in the culture. In conditions with LPS (inducer of the M1 profile) markers CD38 and CD86 should be expressed whereas with IL4 (inducer of the M2 profile) markers CD163 and CD206 should be expressed.
In cytometry, the positive controls are important to validate the results found in the conditions being evaluated. Moreover, we corrected the number sample size in the figure 3 legend. This error happened due to the exclusion of some outliers from some analyses. We apologize for this mistake.
- Additional major limitations of this study looking only at PBMC should be noted in Discussion.
R: We pointed the study limitation, including, in the discussion the sample type (whole blood or PBMC) together with the description of the results.
7.Accurate read-through and extensive set of language corrections is needed to avoid the use of incorrect terms or words (e.g., "in all disease degrees" on lines 119-120 instead of "disease stages" or "levels of disease progression" or, similarly, "disease graveness" on line 131 and in many other places instead of "disease severity" or "so called then as unclassified", line 141 instead of " and are therefore listed as unclassified" or " telomere length was performed", line 159, instead of "was measured"; there are many, many more examples of this).
R: Language corrections have been made and a new revision was carried out in the manuscript. All of these corrections are flagged in red throughout the text in the article.
Round 2
Reviewer 1 Report
The authors have addressed my concerns.